# The Effect of Different Arc Currents on the Microstructure and Tribological Behaviors of Cu Particle Composite Coating Synthesized on GCr15 Steel by PTA Surface Alloying

**Yibo Xiong, Dongqing Lin, Zhizhen Zheng \*, Jianjun Li and Tiantian Deng**

State Key Laboratory of Materials Processing and Die & Mould Technology, College of Materials Science and Engineering, Huazhong University of Science and Technology, Wuhan 430074, China; yiboxiong@hust.edu.cn (Y.X.); Lnpin@hust.edu.cn (D.L.); jianjun@mail.hust.edu.cn (J.L.); dengtt@hust.edu.cn (T.D.)

\* Correspondence: zzz@mail.hust.edu.cn; Tel.: +86-13618660011

**Abstract:** In order to improve the tribological performance of the slanted guide pillar, Cu particle reinforced composite coatings were synthesized on the surface of GCr15 steel using the plasma transferred arc (PTA) alloying technique. A systematic experimental investigation was conducted to study the effects of PTA current on the microstructure and microhardness of alloyed coatings. In addition, tribological behaviors at room temperature (RT) and high temperature (HT) were investigated. The results indicate that at low PTA current (70A), due to the insufficient current, no Cu particles are dissolved in the alloyed coating and a Cu-rich layer is observed on the surface. With the increase in the PTA current, Cu particles are gradually dissolved into the alloyed layer and the microstructure of alloyed coating mainly consists of bamboo-like martensite, retained austenite, and dispersed Cu particles. The microhardness of the PTA samples is approximately four times that of the untreated sample. The tribological results exhibit that an abrasive wear at RT and slight abrasive wear with oxidation wear at HT are the dominant wear mechanisms of alloyed coatings. The PTA samples show far superior antifriction properties compared to the untreated and remolten samples at both RT and HT, which can be attributed to the formation of lubricating Cu films and the improvement in microhardness.

**Keywords:** plasma transferred arc alloying; microstructure; tribological behavior; GCr15 steel; Cu film

## 1. Introduction

As the guide device of the core-pulling mechanism in the injection mold, the performance of the slanted guide pillar has an important influence on the smooth opening and closing of the injection mold. GCr15 steel is an attractive material for the slanted guide pillar because of its high fatigue resistance and good dimensional stability [1]. However, due to the large lateral force and thermal stress during operation, the slanted guide pillar is prone to wear and even breakage [2]. Thus, it is very importance to improve its wear resistance and reduce its friction factor [3].

The methods used in surface modification techniques for coating include thermal spray coating [4], gas tungsten arc welding [5], laser cladding [6], and plasma transferred arc (PTA) treatment [7]. The PTA technique has been widely used to improve the wear resistance and hardness of metallic materials by producing metal matrix composites due to its remarkable advantages, such as good surface

quality, release of the pre-treatment or vacuum environment, high energy efficiency, and competitive running costs [8–10].

Bourithis et al. [11] focused on producing a metal matrix composite tool steel with TiC as reinforcing particles using PTA technology and investigated the wear behavior under different sliding speeds. They demonstrated that the wear rate is 10–100 times greater and the dominant wear mechanism for low sliding speeds is plastic deformation, while at higher sliding speeds an oxidation mechanism predominates. Liu et al. [12] conducted a study on the fabrication of a $TiN(Ti_2N)/Ti$ composite layer using PTA on C17200 alloy, and successfully obtained a better combination of wear resistance and conductive performance. Cao et al. [13] found that synthesizing a high-vanadium high-speed steal alloying layer on nodular cast iron could increase the hardness and improve wear resistance. However, this may not be an effective method for producing a composite layer with hard phases for materials with a high load and high frequency of use, such as slanted guide pillars. Introducing a self-lubricating phase is an effective way to obtain a longer service life.

Cu is an excellent solid lubricant with good thermal conductivity, a low friction coefficient, and good friction compatibility [14–17]. Zeng et al. [18] synthesized a composite coating of Cu on high nickel austenitic ductile iron using laser surface alloying under two different processing conditions. They studied the evolution of the microstructure, distribution of the alloying elements, and hardness of the alloyed zone. However, the mechanical and tribological performance of Cu-Fe composite coatings synthesized using the PTA process has been little known until now.

Therefore, in the present work, composite coatings of Cu were synthesized on the surface of GCr15 steel using PTA technology under different PTA currents. The effects of different PTA currents on the microstructure and microhardness of Cu composite coatings were investigated. The tribological properties of the alloyed coatings were analyzed using dry wear tests at room temperature (298 K) and high temperature (473 K). In addition, an untreated GCr15 sample and a remolten sample without copper addition were used for comparison purposes.

## 2. Experimental Procedures

### 2.1. Material Preparation

GCr15 steel was selected as the substrate material. As shown in Figure 1, the matrix structure is mainly characterized by spherical carbides distributing evenly on the ferrite. The chemical composition of GCr15 steel is shown in detail in Table 1. Specimens with a dimension of 140 mm × 70 mm × 8 mm were cut for the plasma transferred arc (PTA) alloying test.

Copper powder with a purity of 99.99%, as the alloying additive material, was sprayed onto the GCr15 steel surface with a thickness of 80 μm using the thermal spraying technique. For the purpose of obtaining a perfect alloyed coating, the particle size of the copper powder selected was approximately 90 μm. Prior to the thermal spaying, the steel surface was cleaned using acetone and blasted with alumina to improve the bonding strength of the sprayed coating. The specimens were then fixed onto a working platform and scanned using a PTA torch. The PTA alloying process parameters are summarized in Table 2. The remolten sample without added copper has the same PTA parameters as the second sample in Table 2.

**Table 1.** Chemical composition of GCr15 steel (wt.%).

| C | Si | Mn | S | Cr | P | Fe |
|---|---|---|---|---|---|---|
| 0.95–1.10 | 0.15–0.35 | <0.50 | <0.025 | 0.30–1.60 | <0.025 | Bal. |

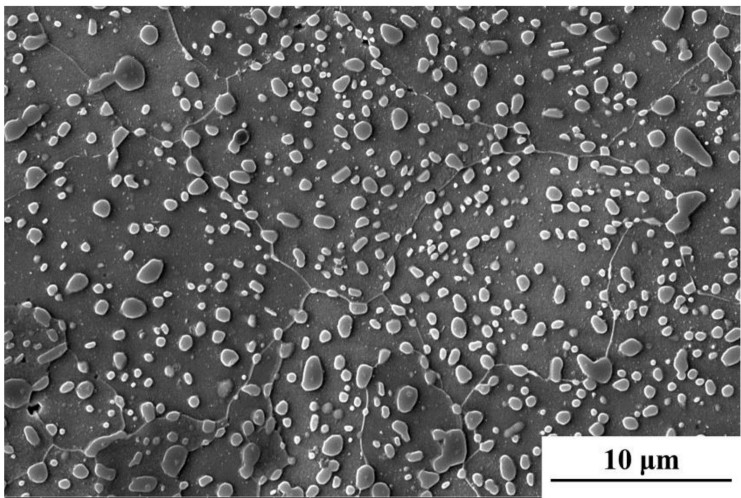

**Figure 1.** Microstructure of the GCr15 steel substrate.

**Table 2.** Plasma transferred arc alloying process parameters.

| Sample No. | PTA Current (A) | Scanning Speed (mm/min) | Gas flow (L/min) | Working Distance (mm) |
|---|---|---|---|---|
| 1 | 70 | 300 | 3 | 5 |
| 2 | 90 | 300 | 3 | 5 |
| 3 | 110 | 300 | 3 | 5 |

*2.2. Microstructure and Microhardness Measurements*

After the PTA treatment, micro metallographic samples in the size of 15 mm × 10 mm × 8 mm were obtained. The samples were mechanically grinded using abrasive papers, and then polished to 0.05 μm with silica and corroded. The etchant was a 4% nitric acid alcohol solution, and the corrosion time was 60 s. The molten pool morphology of the alloyed layer was observed using an optical microscope (OM, VHX-1000C, Keyence, Osaka, Japan). The microstructure was inspected using a Scanning Electron Microscope (SEM, FEI Quanta 200, FEI, Hillsboro, OR, USA) equipped with an X-ray energy dispersive spectrometer (EDS, FEI Quanta 200, FEI, Hillsboro, OR, USA). The different phases presented in the microstructure of the alloyed coating were determined using X-ray diffraction (XRD, XRD-7000, Shimadzu, Tokyo, Japan) with the use of Cu Kα radiation over a range of 2θ from 20° to 90°.

Microhardness was examined using a Vickers hardness tester (DHV-1000, Warwick Technology Ltd., Beijing, China) with an applied load of 4.9 N for a dwell time of 15 s. The first indentation was located at 0.1 mm from the coating surface, and subsequent indentations were separated by 0.1 mm intervals in the cross-section. Each point at the same depth was measured repeatedly three times and an average value was taken for analysis.

*2.3. Wear Test*

To evaluate the tribological properties of the PTA alloyed layer, wear tests were conducted utilizing a ball-on-plate friction wear tester (UMT-Tribolab, Bruker Corporation, Karlsruhe, Germany) which is capable of monitoring and recording the coefficient of friction (COF) automatically and continuously. Wear tests were carried out at room temperature (298 K) and high temperature (473 K), respectively. All the dry sliding wear tests were conducted against a $Si_3N_4$ ball (with a hardness of 98 HRC and a diameter of 6.35 mm) at a constant linear speed of 10 mm/s, with a load of 100 N for the sliding time of 30 min. The wear mass loss of the specimen after each test was weighed using an analytical

balance (AT21, Mettler Toledo, Switzerland) with a weight accuracy of 0.1 mg. The wear rate (W) was calculated in accordance with Equation (1) [19]:

$$W = \frac{\Delta W}{vtF} \tag{1}$$

where $\Delta W$ is the wear mass loss, $v$ is the sliding speed, $t$ is the sliding time, and $F$ is the applied load. The worn morphology was investigated using SEM to reveal the wear mechanisms

## 3. Results and Discussion

### 3.1. Microstructure

Figure 2 shows the cross-sectional alloyed layer distribution of the composite coating under different arc currents. As shown in Figure 2a, at an arc current of 70 A, the molten pool morphology of sample 1 was separated into two regions: the heat affected zone (HAZ), and the substrate zone (SZ). This is the result of there being insufficient energy for the sample to melt the copper into the substrate pool to form an alloyed zone. However, as the arc current increases (90 A, 110 A), it can be clearly seen from Figure 2b,c that the cross-section layers of samples 2 and 3 are divided into three moon-like regions: the alloyed zone (AZ), the heat affected zone (HAZ), and the substrate zone (SZ).

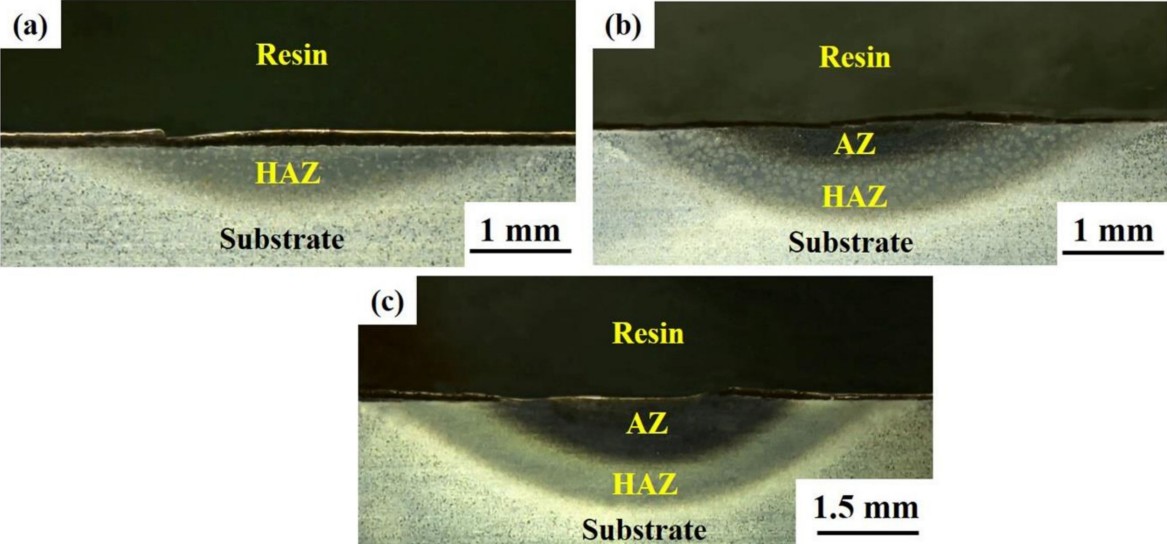

**Figure 2.** The cross-sectional morphology of the molten pool at different PTA currents: (**a**) 70 A; (**b**) 90 A; and (**c**) 110 A.

Furthermore, the arc current also affects the dimensions of the molten pool. Figure 3 lists the dimensions of AZ and HAZ formed at different PTA currents. The comparison of these values shows that the widths and depths of AZ and HAZ increase with the increasing arc current. When the PTA current is 70 A, no obvious alloyed zone is found. The width and depth of the heat affected zone at 70 A are only 4321 mm and 512 mm, respectively. When the PTA current increases to 110 A, the width and depth of the alloyed zone reach 4674 mm and 940 mm, respectively. In addition, the width of the heat affected zone at 110 A reaches 8378 mm, which is almost twice as wide as that at 70 A. The depth of the heat affected zone at 110 A reaches 2218 mm, which is nearly four times as deep as that at 110 A. As the arc current increases, the energy density of the plasma arc in the per unit area grows, inducing a more intense fluctuation of heat flow inside the molten pool, and finally resulting in an increase in the dimensions of AZ and HAZ.

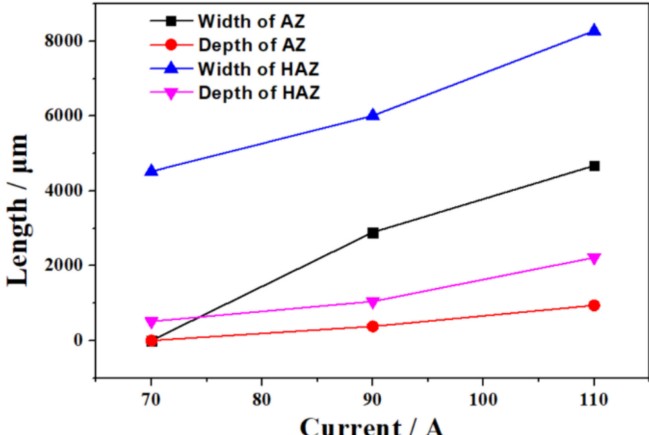

**Figure 3.** The dimensions of AZ and HAZ formed at different PTA currents.

Figure 4 shows the microstructure changes of alloying samples produced under different PTA currents. As shown in Figure 4a, the microstructure of sample 1 at a PTA current of 70 A is mainly composed of fine cryptocrystalline martensite and dispersed granular carbides. There is a white layer on the surface, which is a rich Cu layer. Due to the insufficient energy at the current of 70 A, only a rich Cu layer is formed on the surface of the sample, while no Cu particles are found inside the matrix. Figure 4b,c indicate that the microstructures of AZ consist of the bamboo-like martensite and dispersed spherical Cu particles. The thickness of the rich Cu layer decreases evidently. Compared to Figure 4b, the size of the Cu particles and bamboo-like martensite in Figure 4c is much larger and the amount of Cu particles is also increased. This phenomenon can be explained by the fact that the increase in the energy input results in the martensite becoming relatively strong and the dissolution of Cu becoming more sufficient. Table 3 shows the composition analysis results of the alloyed zones in samples 1 to 3. As shown in Table 3, with the increase of the PTA current, the content of copper in the alloyed zone increase. When the PTA current reaches 110 A, the alloyed zone of the sample 3 has the highest copper content, at 12.89%.

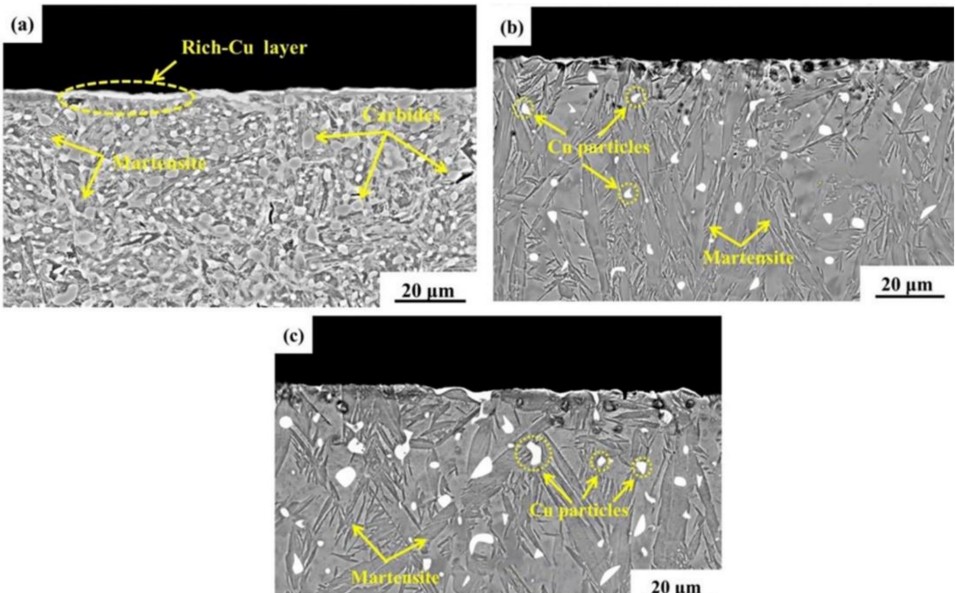

**Figure 4.** SEM microscopic images of microstructure at different PTA currents: (**a**) 70 A; (**b**) 90 A; and (**c**) 110 A.

**Table 3.** Compositions of the alloyed zones in samples 1 to 3 (wt.%).

| Sample No. | Cu | Fe |
|---|---|---|
| 1—70 A | 0 | 100 |
| 2—90 A | 9.75 | 90.25 |
| 3—110 A | 12.89 | 87.21 |

When the PTA current is low, the energy absorbed by the substrate molten pool is less. Therefore, the average temperature of the molten pool is relatively low, and it results in a lower miscibility between the Cu melt and the Fe-substrate melt. For most alloy melts, the viscosity of the alloy melt has a negative relationship with temperature. It means that lower temperature of molten pool will result in greater interfacial tension and higher melt viscosity between the Cu melt and the Fe melt in molten pool. In this situation, only a few of Cu particles are introduced into the Fe-substrate melt zone and there is a Cu-rich layer formed on the surface, such as the sample 1.

As the PTA current increase, more energy is absorbed by the Fe-substrate melt and the average temperature of the molten pool reaches higher. Higher melt pool temperature reduces melt viscosity and interfacial tension. In addition, under the intense convection, the melts in the molten pool are stirred and circulating. This circular stirring motion not only facilitates the full dissolution of the Cu melt, but also enables the Cu element to be evenly dispersed in the Fe molten pool. Finally, it forms the homogeneous Fe-Cu composite molten pool, such as the samples 2 and 3.

The microstructure and EDS analysis of the Fe-Cu alloyed layer are shown in Figure 5. Figure 5a reveals a typical microstructure of the Fe-substrate, which is composed of bamboo-like martensite, and retained austenite. Moreover, there are many white particles distributed in the Fe-substrate. The EDS analysis (Figure 5b) of the white particle denoted as 1 shows that it is rich in Cu. Because of the existence of the liquid miscibility gap in the Cu-Fe system [20] and the high cooling rate of the PTA process [21], liquid phase separation will occur during the PTA alloying process. Therefore, the droplets of the Cu-rich liquid phase are first solidified and kept in a spherical shape in a molten state. At the end of the solidification, the Cu-rich spheres will be distributed on the Fe-substrate.

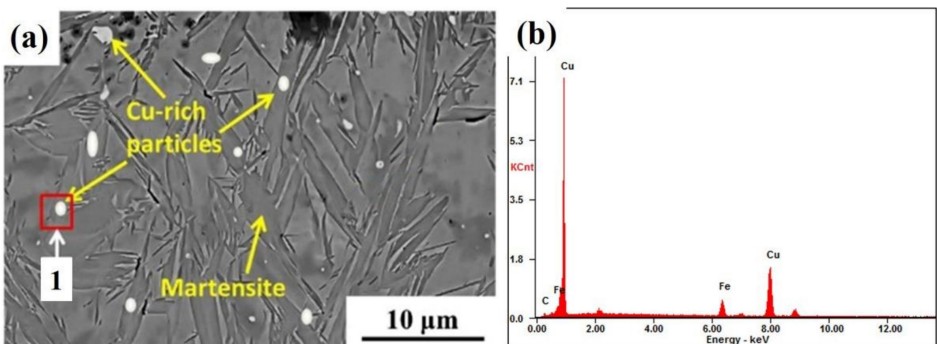

**Figure 5.** SEM micrographs of: (**a**) the alloyed layer; (**b**) the EDS of the Cu-rich particles denoted as 1.

An X-ray diffraction (XRD) analysis was performed in the PTA alloyed samples and the untreated sample. Figure 6 represents the X-ray patterns of the phases existing in the alloyed specimens. Only the $\alpha$-Fe (ferrite) peaks were examined in the untreated sample. In contrast, no diffraction peak of $\alpha$-Fe was found in samples 1, 2 and 3. After the PTA alloying treatment, strong Cu peaks and weak martensite peaks are observed in the diffraction pattern of the PTA sample 1. This result confirms the fact that there is a Cu-rich layer on the surface of sample 1 and that the rich Fe particles are formed in it. The diffraction pattern of the remolten sample presents strong martensite peaks and weak $\gamma$-Fe (retained austenite) peaks. The martensite peaks of this sample are more intense. In samples 2 and 3, as can be expected from the above SEM and EDS analyses, the diffraction peaks of martensite, Cu, and $\gamma$-Fe (retained austenite) were detected. The results of the XRD are consistent with the above results of the microstructure observations.

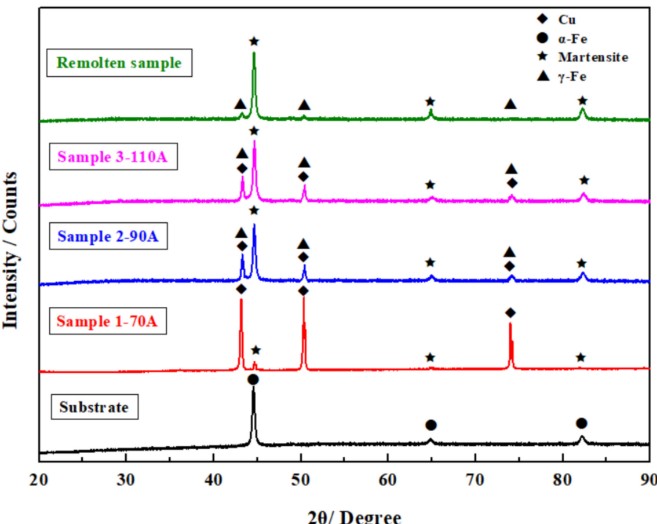

**Figure 6.** X-ray diffraction patterns of the untreated sample, sample 1, sample 2, and sample 3.

### 3.2. Microhardness

Figure 7 represents the microhardness value distribution of the different samples along the depth profiles from surface to substrate. As can be seen from Figure 7, the untreated sample was in the annealed condition, so its microhardness was relatively low. The microhardness of the composite coating was approximately four times greater than that of the untreated substrate. This significant increase in hardness may be related to the existence of large amounts of the hard phases (carbides and martensite). In sample 1, the microhardness near the surface was quite low, which was 85 $HV_{0.5}$, owing to the existence of the Cu-rich layer. After arriving at the alloyed layer, a sharp increase could be seen in the value of microhardness to the highest hardness, at approximately 779.4 $HV_{0.5}$. Then, the value of microhardness decreased gradually with increasing depth and leveled off at approximately 200 $HV_{0.5}$, which was the initial microhardness of the substrate. In the heat affected zone, the microhardness decreases due to the enlarged size of martensite and the increase of the retained austenite. As for samples 2 and 3, since no rich Cu layer was formed or the area of the rich Cu layer was small, there was a slight decrease in the microhardness near the surface. In the AZ of samples 2 and 3, the highest microhardnesses were about 782.2 $HV_{0.5}$ and 793.6 $HV_{0.5}$, respectively. However, the microhardnesses of the PTA alloying samples were slightly lower than that of the remolten sample, the highest microhardness of which was about 856.6 $HV_{0.5}$. This is because Cu particles that are soft phase are introduced in the PTA alloying samples. In addition, the remolten sample contains a higher content of martensite, which can also increase the microhardness of the remolten sample. Although the introduction of Cu particles would slightly reduce the microhardness of the PTA alloying samples, they were still much higher than that of the untreated sample, which was about 200 $HV_{0.5}$.

It can be further observed that with the increase in PTA current, the depth of the high microhardness area of the PTA alloying samples increases. Due to the fact that the heat output of the plasma beam grows as the PTA current increases, the depth of the AZ and HAZ of the PTA alloying samples would also increase.

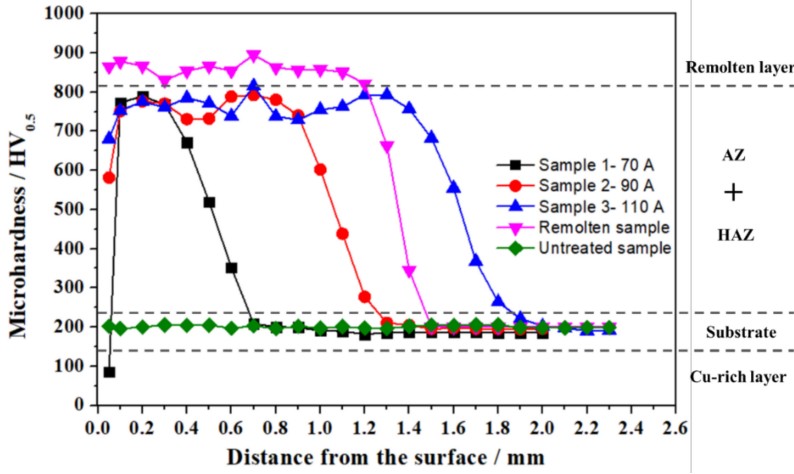

**Figure 7.** The microhardness profiles of the distance from surface.

*3.3. Friction Coefficient and Wear Rates*

Figure 8 presents the variations in the coefficient of friction (COF) curves for the PTA alloying samples, untreated samples, and remolten samples at room temperature (298 K) and high temperature (473 K) in the dry sliding wear test.

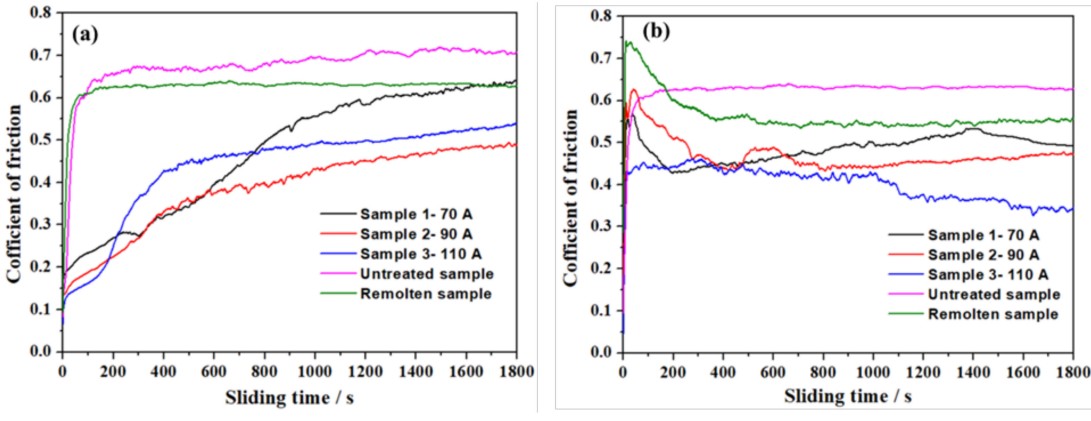

**Figure 8.** The friction coefficient curves of PTA alloying samples, untreated samples, and remolten samples: (**a**) room temperature (293 K); (**b**) high temperature (473 K).

As shown in Figure 8a, at room temperature (293 K), the average coefficients of friction of the untreated sample, the remolten sample, and PTA alloying samples 1, 2, and 3 are 0.675, 0.625, 0.514, 0.446, and 0.459, respectively. It is noteworthy that the average coefficients of friction of the PTA alloying samples are all lower than those of the untreated sample and remolten sample at room temperature. This indicates that after PTA alloying, the dissolution of Cu particles in the samples has a good influence on antifriction. At the beginning of the wear test, the sample 1 shows a low coefficient of friction, because there is a layer of Cu-rich layer on the surface, which plays the role of lubrication. However, as the wear test continued, the coefficient of friction of the sample 1 increases rapidly, gradually exceeding the coefficient of friction of the samples 2 and 3. The reason is that the Cu-rich layer only exists on the surface of the sample 1 and its bonding strength with the Fe-substrate is relatively low, so it is worn gradually during the wear test, leading to the rise of the coefficient of friction. For the samples 2 and 3, Cu particles are embedded in the Fe-substrate. In the process of the wear test, Cu particles are gradually exposed to the surface and formed copper film, so the coefficients of friction of the samples 2 and 3 go into the steady stage after the running-in stage. As for PTA alloying sample 2, the martensite and the Cu particles in the alloying layer are smaller and distributed uniformly, hence it has the lowest COF and the best antifriction effect. Figure 8b shows the

coefficients of friction of the untreated sample, the remolten sample, and the PTA alloying samples at high temperature (473 K). As can be seen from the picture, the average coefficients of friction of the untreated sample, the remolten sample, and PTA alloying samples 1, 2, and 3 are 0.638, 0.588, 0.502, 0.441, and 0.409, respectively. The coefficients of friction of all the samples are lower at high temperature (473 K) than room temperature (293 K). The COFs of all the PTA alloying samples are also much lower than those of the untreated sample and remolten sample at 473K. In addition, PTA alloying sample 3 shows the lowest COF. This is due to the fact that the wear mechanism at high temperature has changed, and the oxidation wear is the main factor at high temperature.

Figure 9 presents a comparison of the wear rates among the PTA alloying samples, the untreated samples, and the remolten samples at room temperature (RT) and high temperature (HT). It can be noted that the wear rates of the PTA alloying samples and the remolten samples are far lower than those of the untreated samples at both RT and HT, demonstrating that wear resistance can be significantly improved by the PTA process. It can be clearly observed that, at room temperature, the lowest wear rate of the PTA alloying samples is obtained at the PTA current of 90 A ($\sim 4.78 \times 10^{-8} g/Nm$). Furthermore, the wear rate of the remolten sample ($\sim 5.45 \times 10^{-8} \ g/Nm$) is almost the same as that of the PTA samples, but much lower than that of the untreated sample ($\sim 3.86 \times 10^{-6} \ g/Nm$). In addition, it is worth noting that the specific wear rates of all the samples at HT are greater than at RT. In addition, at high temperature, the minimum wear rate is obtained with the remolten sample ($\sim 1.05 \times 10^{-7}$ g/Nm). The wear rate of the PTA sample at 110 A ($\sim 1.09 \times 10^{-7}$ g/Nm) is slightly higher than that of the remolten sample at 473K, which is much lower than that of the untreated sample ($\sim 4.83 \times 10^{-6}$ g/Nm).

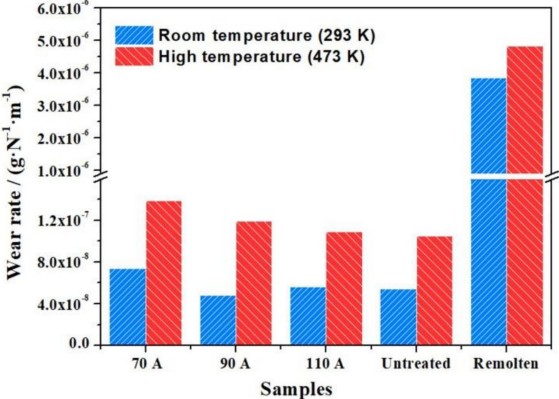

**Figure 9.** The wear rates of PTA alloying samples, untreated samples, and remolten samples.

By comparing Figures 8 and 9, the PTA alloying samples at RT and HT present lower COFs and wear rates than those of the untreated samples in dry sliding condition. The PTA alloying samples also show lower COFs than those of the remolten samples and similar wear rates to those of the remolten samples at RT and HT in dry sliding conditions. Such a reduction in COF and wear rates may be attributed to the lubricating effect of the copper addition and the increase in microhardness.

### 3.4. Worn Surface Analysis and Wear Mechanisms

### 3.4.1. Worn Surfaces at Room Temperature

The worn surface morphologies of the PTA alloying samples, the untreated sample, and the remolten sample after the dry sliding wear test at RT are shown in Figure 10. It can be clearly observed from Figure 10a that severe damage with extensive plastic deformation and delamination is generated, which means that adhesive wear dominates the wear mechanism of the untreated sample at room temperature. This result can be explained by the fact that the untreated sample (200 HV) is much softer than the $Si_3N_4$ counter ball (1540 HV), which means that the surface is easily ploughed or

cut by the hard counterpart. In Figure 10b, as the surface of the remolten sample is quenched to form a hard phase (martensite), which plays a supporting role in the friction process, a large amount of ploughing appears on the wear surface. In addition, due to the local high temperature on the surface, adhesive scratches occur, resulting in metal fatigue shedding. Therefore, some intense ploughing and delamination can be observed in the remolten sample, revealing a combination of adhesive wear and abrasive wear.

From the worn surface morphology of the PTA alloying sample at the arc current of 70 A shown in Figure 10c, some slight ploughing and adhering wear can be observed, which is similar to the worn morphology of the remolten sample due to the similar microstructure. However, there is a rich Cu layer on the surface of sample 1. It serves a lubrication function, which can achieve the effect of antifriction, thus the degree of wear is relatively low. The SEM micrograph in Figure 10d indicates that the worn surface of the alloying sample at 90 A is smooth with some mild scratches and shallow grooves. There are some tiny metallic debris appeared on the worn surface, which indicates that slight abrasive wear [22] occurred in the sliding wear test. These tiny particles between the sliding surfaces would produce a grinding action during the sliding process, making the worn surface smooth. The similar but slightly severe worn surface morphology can be observed in Figure 10e. This is due to the fact that the surface roughness of the sample increases due to the excessive current, and the degree of wear increases.

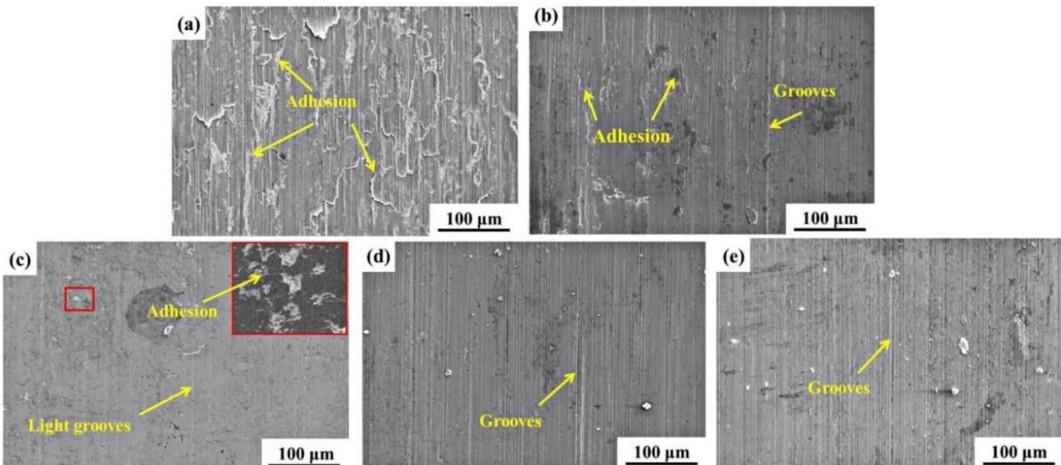

**Figure 10.** SEM micrographs of worn surface at RT: (**a**) the untreated sample; (**b**) the remolten sample; (**c**) the PTA alloying sample at 70 A; (**d**) the PTA alloying sample at 90 A; (**e**) the PTA alloying sample at 110 A.

After PTA alloying, a large amount of copper is dissolved into the alloying layer. Cu is soft in texture and ductility. In the process of the wear test, it gradually spreads across the metal surface and forms a Cu lubricating film. At this point, friction induces adhesive wear and abrasive wear mainly occurs in the copper film. As the shear strength of Cu is smaller than that of steel, the friction and COF become smaller, and the wear morphology is also relatively smooth. Moreover, due to its good thermal conductivity, Cu can reduce the friction surface temperature and further reduce adhesive wear. In summary, the tribological properties of the PTA alloying samples at room temperature are better than those of the untreated sample and the remolten sample. The tribological property of the PTA alloying sample at 90 A is the best of all the alloying samples.

### 3.4.2. Worn Surfaces at High Temperature

Figure 11 represents the SEM and Back Scattered Electron (BSE) micrographs of the worn surfaces of different samples at high temperature. As shown in Figure 11a1, no significant difference in the worn surface of the untreated sample can be observed between the wear test processed at RT and HT.

In Figure 11a2, there is an oxide layer on the surface, which implies that the wear mechanism of the untreated sample at high temperature is oxidation wear and adhesive wear. Figures 11 and 11 show that the worn surface of the remolten sample presents a few traces of adhesion, many parallel grooves, and an oxide layer, indicating that the wear mechanism of the remolten sample turns to abrasive wear, adhesive wear, and oxidation wear. As can be seen from Figure 11c1–e2, the main wear mechanism of the PTA alloying samples is slight abrasive wear and oxidation wear. As the PTA current increases, the worn surface becomes increasingly smooth, and the grooves of the surface become increasingly shallow. This is due to the fact that the dissolution of Cu particles becomes more sufficient and a larger lubricating Cu film is formed with the increase in the PTA current. At a PTA current of 110 A, some Cu particles are gradually exposed to the surface during the sliding process and these Cu particles are smeared by the hard friction pair after plastic deformation, which leads to the formation of the lubricating Cu film.

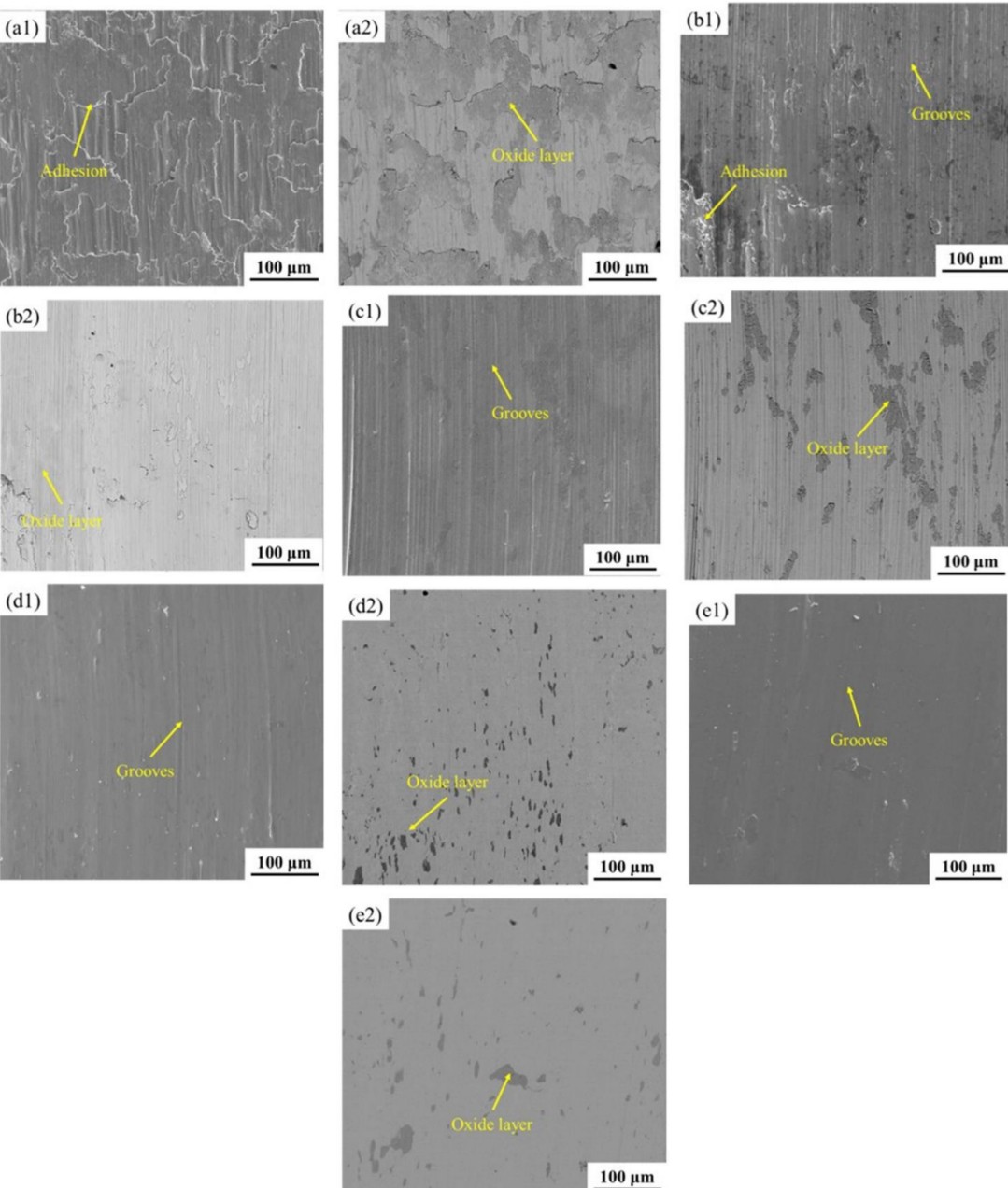

**Figure 11.** SEM and BSE micrographs of the worn surface at HT: (**a1**,**a2**) untreated sample; (**b1**,**b2**) remolten sample; (**c1**,**c2**) PTA alloying sample at 70 A; (**d1**,**d2**) PTA alloying sample at 90 A; (**e1**,**e2**) PTA alloying sample at 110 A.

### 3.4.3. Wear Mechanism

In order to further explore the influences of Cu particles on tribological properties and the wear mechanism, a schematic representation of the entire tribological process is shown in Figure 12. During the initial stage of the wear process, a small amount of wear debris is identified, as shown in Figure 12a; this stage is the asperity contact stage, during which surface contact only occurs between the asperities and causes the generation of wear debris [23]. While the Cu particles penetrated in the alloyed layer of the substrate are still buried, wear sliding only occurs on the top surface of the substrate. As the wear process advances, hard asperities tend to plough the matrix deeper and more debris is generated from the fracture of the asperities. Due to the high temperature and chemical composition from the atmosphere involved, oxidation of the debris and alloyed surfaces takes place (Figure 12b). At this stage, Cu particles begin to be exposed on the friction surface, while the Fe matrix on the top of the surface is worn off. As the wear process continues, oxides and the debris are further rubbed and spread out. The Cu plastic layer is also further ground, smeared, and as a result forms a thin Cu film on the surface of the composite coating (Figure 12c). Once the Cu film is formed on the coating surface, further sliding is expected to occur between additional tribolayers. The Cu film on the sliding surface can bear the friction force and prevent wear loss. In this way, it avoids direct friction contact and contributes to the good antifriction performance. Ezirmik et al. [24] and Wei et al. [25] have confirmed that Cu addition can significantly enhance tribological properties, acting as a solid lubricant. The formation of the plasticized Cu film formed between surfaces can also protect contacting surfaces from extreme wear and effectively improve wear resistance. Therefore, it can be concluded that the self-lubricating effect of Cu particles weakens the adhesion of the sliding surface and reduces the friction force between the friction part and friction pair. Thus, only light abrasive wear and oxidative wear occurred in the present study.

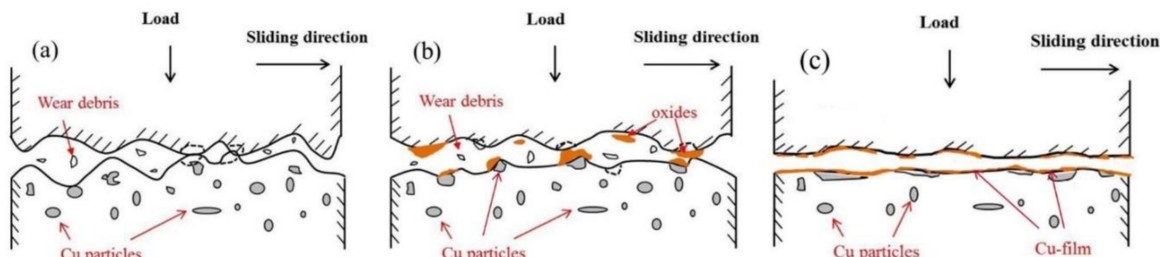

**Figure 12.** Schematic representation of the entire tribological process.

## 4. Conclusions

In this paper, the plasma transferred arc (PTA) alloying surface method was successfully used to synthesize Cu composite coatings on the GCr15 steel surface. The effects of the arc current on the microstructures of the coatings were investigated, and the microhardness and wear behavior of the as-received and Cu coating alloys were determined. The following conclusions can be made based on the results obtained:

(1) The microstructures of the alloyed coatings are distinctly influenced by the arc current. After PTA alloying at 90 A and 110 A, the alloyed coating is composed of bamboo-like martensite, retained austenite, and Cu-rich particles. Cu-rich particles are gradually embedded in the substrate because of the liquid phase separation and high cooling rate of the PTA process.

(2) The microhardness of the alloyed coatings is significantly improved and is approximately four times higher than that of the untreated sample. However, it is slightly lower than that of the remolten sample, due to the dissolution of the Cu particles. With the increase in the PTA current, the location of the maximum microhardness of the PTA alloying samples becomes further away from the surface.

(3)   The Cu composite coatings exhibit outstanding antifriction behavior. Compared to the untreated sample, the COFs and the wear rates of the PTA alloying samples are significantly lower due to the lubricating effect of Cu particles.

(4)   The dominant wear mechanism of the alloyed coatings at RT is slight abrasive wear, while the rise in the test temperature leads to a shift in the wear mechanism to mild abrasive wear and oxidative wear at HT. The Cu-rich particles contained in the alloyed layer can form Cu films which have a lubricating effect and prevent the alloyed coating from undergoing severe wear.

The results confirm that the PTA surface alloying of Cu particle can be used to improve the tribological performance of GCr15 steel and that this Cu composite coating shows lower coefficient of friction and superior antifriction properties compared to the untreated sample. In order to apply this technique to the real guide pillars, reasonable path planning of PTA surfacing alloying seems worth investigating further. Also, more detailed investigations of the microstructural evolution and performance evolution during PTA double treatment are needed as the main direction for future work.

**Author Contributions:** Y.X. performed the data analyses, collected the literature and wrote the manuscript; D.L. and T.D. contributed significantly to execution of experiments and data collection; Z.Z. contributed to the conception of the study and design of the experiment; J.L. helped perform the analysis with constructive discussions.

**Funding:** This research was funded by the National Natural Science Foundation of China (Grant No. 51435007).

**Acknowledgments:** The authors would like to express their great appreciation for the kind help of the Analysis and Test Center of HUST (Huazhong University of Science and Technology). The financial support of the State Key Laboratory of Materials Processing and Die & Mould Technology is also gratefully acknowledged.

**Conflicts of Interest:** The authors declare no conflict of interest.

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
