# Peer review of "The Effect of Different Arc Currents on the Microstructure and Tribological Behaviors of Cu Particle Composite Coating Synthesized on GCr15 Steel by PTA Surface Alloying"

_metals, doi:10.3390/met8120984_

Round 1

Reviewer 1 Report

This is an interesting paper, with a solid experimental approach, a complete discussion and consistent conclusions.

Some important comments apply:

- Please provide evidence and ease of identification of the type of steel used for the substrate also for non-Chinese readers, by either using the term "bearing steel" or reporting equivalent grades (such as 100Cr6 or ASTM grade).

- Fig. 2 is not necessary and can be omitted.

- p.5, line 135 and Fig. 5a: what is the evidence of the presence of retained austenite in the remolten layer?

- p.6 line 170 and Fig.7: why wasn't the remolten substrate characterized by XRD? This could have helped to resolve and discriminate the Cu peaks from those of austenite.

- p.6, lines 143-159: the discussion of Cu and Fe particles/layers/liquid phase is very confusing. It is not clear what Authors mean by "the composite coatings of the samples can be further divided into rich Cu layer and rich Fe layer". Also the sentence "The Cu-rich layer had the same origin of Cu-rich liquid phase, which is solidified later than the Fe-rich liquid phase" is not clear. Authors are invited to completely re-write this part of the discussion.

- Fig. 8: a Vickers hardness of 200 seems too low for a bearing steel. Authors are invited to comment on that.

- p.10, line268: the term "Cu protective film" is not correct, and should rather be replaced with "Cu lubricating film".

- It would be very important to comment, in conclusion the paper, on the ease of surface PTA treatment of real shape components, offering an indication of expected uniformity of the remelting treatment on the whole surface of the guide pillars, and on the potential effect of PTA double treatment in the border areas.

- English language needs substantial revision.

Author Response

Response to Reviewer 1 Comments

Point 1: Please provide evidence and ease of identification of the type of steel used for the substrate also for non-Chinese readers, by either using the term "bearing steel" or reporting equivalent grades (such as 100Cr6 or ASTM grade).

Response 1: Thank you for your suggestion. SUJ2 is one of the bearing steel of the Japanese JIS brand. We have replaced the term SUJ2 with GCr15, which is a more familiar name. The chemical compositions of SUJ2 steel and GCr15 steel are shown in the following table. As can be seen from the Table, the SUJ2 steel and GCr15 steel have the same chemical composition.

C

Si

Mn

S

Cr

P

Fe

SUJ2

0.95-1.10

0.15-0.35

<0.50

<0.025

0.30-1.60

<0.025

Bal.

GCr15

0.95-1.10

0.15-0.35

<0.50

<0.025

0.30-1.60

<0.025

Bal.

Point 2: Fig. 2 is not necessary and can be omitted.

Response 2: Thank you for your advice. We have removed figure 2.

Point 3: p.5, line 135 and Fig. 5a: what is the evidence of the presence of retained austenite in the remolten layer?

Response 3: According to the XRD results, the diffraction peak of γ-Fe(retained austenite) is detected. Therefore, we think that is the evidence of the presence of retained austenite in the microstructure of the PTA alloying samples. However, it is difficult to find the exact position of retained austenite in the metallographic images, and we did not do some experiments to show the exact position of retained austenite, such as EBSD. So we get rid of the arrows marked with retained austenite in Fig. 5 (in the revised manuscript Fig. 4) and delete the term “retain austenite” in line 135 and line 131.

Point 4: p.6 line 170 and Fig.7: why wasn't the remolten substrate characterized by XRD? This could have helped to resolve and discriminate the Cu peaks from those of austenite.

Response 4: Thank you for your suggestion. We have done the XRD test of the remolten sample. Fig. 7 (in the revised manuscript Fig. 6) has been re-drawn and the XRD analysis of the remolten sample has been added in Line 170. As can be seen from the diffraction pattern of the remolten sample, strong martensite peaks and weak γ-Fe (retained austenite) peaks are detected. This sample lacks of Cu, so no Cu peaks are observed.

Line 170, “The diffraction pattern of the remolten sample presents strong martensite peaks and weak γ-Fe (retained austenite) peaks. The martensite peaks of this sample are more intense.” has been added.

Point 5: p.6, lines 143-159: the discussion of Cu and Fe particles/layers/liquid phase is very confusing. It is not clear what Authors mean by "the composite coatings of the samples can be further divided into rich Cu layer and rich Fe layer". Also the sentence "The Cu-rich layer had the same origin of Cu-rich liquid phase, which is solidified later than the Fe-rich liquid phase" is not clear. Authors are invited to completely re-write this part of the discussion.

Response 5: Thank you for your suggestion. We have re-written this part according to the suggestion.  Fig. 6 (in the revised manuscript Fig. 5) has been updated. The re-written part is as follows:

When the PTA current is low, the energy absorbed by the substrate molten pool is less. Therefore, the average temperature of the molten pool is relatively low, and it results in a lower miscibility between the Cu melt and the Fe-substrate melt. For most alloy melts, the viscosity of the alloy melt has a negative relationship with temperature. It means that lower temperature of molten pool will result in greater interfacial tension and higher melt viscosity between the Cu melt and the Fe melt in molten pool. In this situation, only a few of Cu particles is introduce into the Fe-substrate melt zone and there is a Cu-rich layer formed on the surface, such as the sample 1.

As the PTA current increase, more energy is absorbed by the Fe-substrate melt and the average temperature of the molten pool reaches higher. Higher melt pool temperature reduces melt viscosity and interfacial tension. In addition, under the intense convection, the melts in the molten pool are stirred and circulating. This circular stirring motion not only facilitates the full dissolution of the Cu melt, but also enables the Cu element to be evenly dispersed in the Fe molten pool. Finally, it forms the homogeneous Fe-Cu composite molten pool, such as the sample 2 and the sample 3.

The microstructure and EDS analysis of the Fe-Cu alloyed layer are shown in Fig. 6. Fig. 6a reveals a typical microstructure of the Fe-substrate, which is composed of bamboo-like martensite, and retained austenite. Moreover, there are many white particles distributed in the Fe-substrate. The EDS analysis (Fig. 6b) of the white particle denoted as 1 shows that it is rich in Cu. Because of the existence of the liquid miscibility gap in the Cu-Fe system [20] and the high cooling rate of the PTA process [21], liquid phase separation will occur during the PTA alloying process. Therefore, the droplets of the Cu-rich liquid phase are first solidified and kept in a spherical shape in a molten state. At the end of the solidification, the Cu-rich spheres will be distributed on the Fe-substrate.

Point 6: Fig. 8: a Vickers hardness of 200 seems too low for a bearing steel. Authors are invited to comment on that.

Response 6: In this paper, we used spheroidized annealing GCr15 steel, so its hardness is about 200 HV0.5. The hot rolled microstructure of GCr15 steel is usually composed of lamellar pearlite and a small amount of grain boundary carbides, which has high hardness, low plasticity and poor machining performance. Therefore, the spheroidizing annealing of GCr15 steel is needed to spheroidize carbides before the subsequent processing. After the spheroidizing annealing process, the microstructure of GCr15 steel is characterized by spherical carbides distributing evenly on the ferrite. Although the hardness of GCr15 steel is reduced, the machinability and serviceability are improved.

Line 181, “As can be seen from Fig. 8, the untreated sample was in the annealed condition, so its microhardness was relatively low.” has been added.

Point 7: p.10, line268: the term "Cu protective film" is not correct, and should rather be replaced with "Cu lubricating film".

Response 7: Thanks for your advice. We are very sorry for our incorrect writing. We have replaced the term “Cu protective film” with “Cu lubricating film” in the line 23 and line 268.

Point 8: It would be very important to comment, in conclusion the paper, on the ease of surface PTA treatment of real shape components, offering an indication of expected uniformity of the remelting treatment on the whole surface of the guide pillars, and on the potential effect of PTA double treatment in the border areas.

Response 8: Thank you for your suggestion. We are working hard to apply PTA surface alloying of Cu particle to the real guide pillars, but the results are not ideal. So it seems worth investigating further. In addition, we have investigated some effect of PTA double treatment in the border areas, such as a lower hardness existing in the area between the HAZ of the second alloyed molten pool and the first alloyed molten pool. However, we think that the results are incomplete. So, more detailed investigations of the microstructural evolution and performance evolution during PTA double treatment are the main direction of future work.

Line 368, “The results confirm that PTA surface alloying of Cu particle can be used to improve the tribological performance of GCr15 steel and that this Cu composite coating shows lower coefficient of friction and superior antifriction properties compared to the untreated sample. In order to apply this technique to the real guide pillars, reasonable path planning of PTA surfacing alloying seems worth investigating further. Also, more detailed investigations of the microstructural evolution and performance evolution during PTA double treatment are needed as main direction for future work.” has been added.

Point 9: English language needs substantial revision

Response 9: Thank you for your suggestion. We have revised English language using the MDPI English Editing Service.

Reviewer 2 Report

This manuscript reports a plasma transferred arc (PTA) alloying of Cu on the surface of SUJ2 steel, as a function of current, 70-110A. Results indicate enhancement of the micro hardness and wear resistance of alloys prepared at 110A over non-alloyed, remolten sample, or than samples prepared at lower currents. The results are correlated well with the surface morphology, elemental composition and structure. The manuscript is well-written and results clearly presented. This work should appeal to a broad audience of scientists and engineers working on plasma, metals and surface sciences. I have only one question regarding the results. The manuscript is then recommend publication without further revisions needed.

Question: are elemental composition and crystallinity changes for Fe-rich zone, shown in Figs 5-7, specifically related to the heat affected zone? Make this clear in the text

Author Response

Response to Reviewer 2 Comments

Point 1: Are elemental composition and crystallinity changes for Fe-rich zone, shown in Figs 5-7, specifically related to the heat affected zone? Make this clear in the text.

Response 1: Thank you for your suggestion. We have analysed the elemental composition of the alloyed zones in samples 1 to 3. The following has been added to the text:

Table 3 shows the composition analysis results of the alloyed zones in samples 1 to 3. As shown in Table 3, with the increase of the PTA current, the content of copper in the alloyed zone increase. When the PTA current reaches 110A, the alloyed zone of the sample 3 has the highest copper content, at 12.89%.

Table 3. Compositions of the alloyed zones in samples 1 to 3 (wt.%).

Sample no.

Cu

Fe

 1 - 70A

0

100

 2 – 90A

9.75

90.25

 3 – 110A

12.89

87.21

Reviewer 3 Report

1) Explanation of plating technique.

2) Figures 3 is not clear; alloyed zone?

3) Explanation of fig. 4.

4) Fig. 8 detailed explanation.

5) Fig. 9a; strange results. 

Author Response

Response to Reviewer 2 Comments

Point 1: Explanation of plating technique.

Response 1: Thank you for your suggestion. In this study, we used the thermal spraying technology. We have made the following modifications and supplements in the paper: Copper powder with purity of 99.99%, as the alloying additive material, was sprayed on SUJ2 steel surface with a thickness of 80 μm through the thermal spraying technique. For the purpose of obtaining a perfect alloyed coating, particle size of copper powder was selected about 90 μm. Prior to the thermal spaying, the steel surface was cleaned by acetone and blasted with alumina to improve the bonding strength of sprayed coating.

Point 2: Figures 3 is not clear; alloyed zone?

Response 2: Thank you for your advice. Figure 3 (in the revised manuscript Figure 2) has been redone. The alloyed zone represents the area that Cu particles are dispersed on the Fe substrate due to the intense convection during the PTA process.

Point 3: Explanation of fig. 4.

Response 3: Thank you for your suggestion. Line 123, “When the PTA current is 70A, no obvious alloyed zone is found. The width and depth of the heat affected zone at 70A are only 4321mm and 512mm, respectively. When the PTA current increases to 110A, the width and depth of the alloyed zone reach 4674mm and 940mm, respectively. In addition, the width of the heat affected zone at 110A reaches 8378mm, which is almost twice as wide as that at 70A. The depth of the heat affected zone at 110A reaches 2218mm, which is nearly four times as deep as that at 110A.” has been added.

Point 4: Fig. 8 detailed explanation.

Response 4: Thank you for your suggestion. Line 182, “The microhardness of the composite coating was approximately 4 times greater than that of the untreated substrate. This significant increase in hardness may be related to the existence of large amounts of the hard phases (carbides and martensite).” has been added.

Line 189, “In the heat affected zone, the microhardness decreases due to the enlarged size of martensite and the increase of the retained austenite.” has been added.

Line 201-204, “with the increase of PTA current, the depth of the high microhardness areas of the PTA alloying samples increase. Due to the fact that the heat output of the the plasma beam would grow as the increase of PTA current, the depth of AZ and HAZ of the PTA alloying samples would also increase.” has been corrected.

Point 5: Fig. 9a; strange results.

Response 5: In Fig. 9a (in the revised manuscript Fig. 8a), at the beginning of the wear test, the sample 1 shows a low coefficient of friction, because there is a layer of Cu-rich layer on the surface, which plays the role of lubrication. However, as the wear test continued, the coefficient of friction of the sample 1 increases rapidly, gradually exceeding the coefficient of friction of the samples 2 and 3. The reason is that the Cu-rich layer only exists on the surface of the sample 1 and its bonding strength with the Fe-substrate is relatively low, so it is worn gradually during the wear test, leading to the rise of the coefficient of friction. For the samples 2 and 3, Cu particles are embedded in the Fe-substrate. In the process of the wear test, Cu particles are gradually exposed to the surface and formed copper film, so the coefficients of friction of the samples 2 and 3 go into the steady stage after the running-in stage.

Round 2

Reviewer 3 Report

The manuscript now can be published.